METHODS

# *WormSNAP*: A software for fast, accurate, and unbiased detection of fluorescent puncta in *C. elegans*

Araven Tiroumalechetty[1], Elisa B. Frankel[1,2], Peri T. Kurshan[1]*

**1** Department of Neuroscience, Albert Einstein College of Medicine, Bronx, New York, United States of America, **2** Department of Biology, University of Puget Sound, Tacoma, Washington, United States of America

* peri.kurshan@einsteinmed.edu

## Abstract

The detection and characterization of fluorescent puncta are critical tasks in image analysis pipelines for fluorescence imaging. Existing methods for quantitative characterization of such puncta often suffer from biases and limitations, compromising the reliability and reproducibility of results. Moreover, the widespread adoption of many available analysis scripts is often hampered by over-optimization for specific samples, requiring extensive coding knowledge to repurpose for other datasets. We present *WormSNAP (Worm SyNapse Analysis Program)*, a license-free, stand-alone, no-code approach to automated unbiased detection and characterization of 2D fluorescent puncta, originally developed to characterize images of the synapses residing in *C. elegans* nerve cords but suitable for broader 2D fluorescence image analysis. *WormSNAP* incorporates a local means thresholding algorithm and a user-friendly Graphical User Interface (GUI) for efficient and accurate analysis of large datasets, with user control of thresholding and restriction parameters and visualization options for further refinement. *WormSNAP* also calculates three types of correlation metrics for 2-channel images, enabling users to select the ideal metric for their dataset. *WormSNAP* provides robust and accurate fluorescent puncta detection in a variety of conditions, accelerating the image analysis workflow from data acquisition to figure generation.

## Author summary

Here we describe software designed to increase the ease, speed and reliability of analyzing images of fluorescently tagged proteins in *C. elegans*, a microscopic worm proven to be a powerful model system for uncovering principles of biology. Many proteins accumulate sub-cellularly into clusters that can be seen as puncta within fluorescent images. Changes in the localization or attributes of

**Data availability statement:** All data is available at: https://github.com/KurshanLab/WormSNAP_Data (DOI: doi.org/10.5281/zenodo.15283078).

**Funding:** PK received funding from the NIH (NINDS R01-NS123645), the Mathers Foundation and the McKnight Foundation in support of this work. The funders did not play any role in the study design, data collection and analysis, decision to publish, or preparation of the manuscript.

**Competing interests:** The authors have declared that no competing interests exist.

these puncta can indicate cellular dysfunction. Our lab studies synapses, or the connections between neurons, often by visualizing tagged synaptic proteins. Existing image analysis tools were difficult to use, often requiring coding skills, with limited flexibility in parameters tailored to specific datasets, or were available only through costly licenses. Others offered only narrow analyses often limited to single-channel images. To address these issues, we developed WormSNAP, a free, stand-alone tool for automated and unbiased puncta detection. With its intuitive interface, WormSNAP enables efficient analysis of large datasets, offering adjustable detection parameters, dataset-wide visualization, multi-channel puncta detection and overlays of detected puncta without requiring any coding knowledge. WormSNAP can be broadly applied to a wide variety of one and two-dimensional fluorescent images and we anticipate it will be a valuable resource across many areas of biology.

## Introduction

### Fluorescent puncta detection in *C. elegans* research

*C. elegans* research has progressed in tandem with live fluorescent imaging, beginning with the validation of GFP as a marker for gene expression in eukaryotes [1]. Since then, the genetic tractability of *C. elegans* has enabled the generation of numerous cell-specific fluorescent protein (FP) expression systems, ranging from overexpressed plasmid-based arrays to endogenous CRISPR-based reporters, leading to major discoveries about the molecular basis underlying diverse cellular processes [2]. For example, fluorescence imaging of the *C. elegans* nervous system has yielded significant insights into the roles of synaptic proteins and enabled the dissection of complex cell-biological signaling pathways involved in synapse formation, patterning, maintenance, and plasticity [3–7]. Indeed, detection and characterization of fluorescent puncta in *C. elegans* is often a critical step in understanding the nature and extent of mutant phenotypes, thus shedding light onto the biological implications of various mutations.

Although genetic and microscopy innovations have gradually reduced the barriers to fluorescence image acquisition in the *C. elegans* nervous system, the quantification of fluorescence imaging data remains non-standardized. Segmentation methods for defining fluorescent regions of interest (ROIs) vary widely, and most researchers carry out fluorescent puncta detection manually or through custom scripts, often optimized for a specific reporter, cell morphology, phenotype of interest, or dataset [7,8]. Such strategies can expose findings to experimenter bias and jeopardize reproducibility, as observed in other image analysis contexts [9]. Substantial variation also exists in how commonly reported fluorescence puncta characteristics – such as intensity, distribution, and circularity – are measured and analyzed. Furthermore, most freely available analysis software for fluorescence puncta quantification requires some coding expertise to implement or modify and is infrequently updated or supported, leaving researchers to choose between subpar automated methods or tedious manual annotation.

*C. elegans* exhibit a highly stereotypical body plan and nervous system arrangement (schematized in Fig 1A). Most images of synaptic puncta in *C. elegans* are collected as either widefield fluorescent images or z-stack projections from a confocal microscope. Images of *en passant* synapses of the nerve cords are generally cropped and often straightened with the use of a segmented line tool and straighten line function in Fiji (producing uniform width, cropped images referred to as *crops* in this paper (Fig 1B, 1C) [10]. Straightening of a segmented line involves fitting a cubic spline curve to the points that define the line and then joining the splines together into a straightened line. A 1D sum projection of each *crop* is then used to generate an intensity plot profile for further analysis (Figs 1D, S1) [8,11,12]. However, reducing images to 1D projections discards much of the spatial information about the fluorescent puncta, restricting the output information primarily to intensity along the process. Although some progress has been made in switching to automated identification of 2D regions of interest (ROIs) to retain spatial information [7,12], no widely adopted method yet exists in this subfield. Meanwhile, CRISPR-mediated endogenous protein labeling – while less prone to overexpression artifacts – often results in dimmer signals and, consequently, lower signal-to-noise ratios (Compare Fig 1B to Fig 1C). This further reduces the suitability of intensity profiles for puncta detection and characterization.

## Software overview

To reduce the barrier to access for unbiased automated puncta quantification in the field, we designed a license-free, 'no-code' platform for optimizing quantification parameters while also providing a reliable, one-size-fits-all base methodology for quantification. Our software offers a MATLAB-based Graphical User Interface (GUI) tailormade for the detection and characterization of discrete fluorescent puncta in cropped 2D images. The software is distributed both as a MATLAB app and as a stand-alone application, allowing users to run it without requiring a MATLAB license or coding knowledge. Our GUI, *WormSNAP,* implements three main functions: 1) a user-adjustable local means thresholding algorithm for puncta detection, 2) restriction of ROIs based on several characteristics, and 3) visualization of segmented images and associated ROIs. *WormSNAP* enables the detection of puncta from the typical *en passant* synapses in linear axons (Fig 1D) and in worm nerve cords (S1 Fig), and more complex synapses from outside the nerve cord (S2 Fig).

We have also included several ease-of-use features such as the ability to save previously used settings, dataset exclusions, ROI detection parameters, and ROI mask visualizations. Outputs include calculations of commonly quantified parameters such as correlation metrics for 2D images, montages of all analyzed images, summary plots, and organized data tables for downstream analysis using Graphpad Prism or other plotting software. The WormSNAP software, instructions for its usage, and a series of Fiji macros for generating straightened cropped images from a variety of common imaging file formats are all available on a GitHub repository: github.com/Kurshanlab/WormSNAP.

## Results

*WormSNAP* has been developed to process cropped and straightened images of up to two fluorescent reporters, hereafter simply termed *crops*. The software was developed using MATLAB R2022a with the Image Processing Toolbox version 11.5 and Statistics and Machine Learning Toolbox version 12.3. The software and associated documentation are available as either a MATLAB add-on or a standalone application (which does not require having a MATLAB license) on the GitHub repository: github.com/KurshanLab/WormSNAP. Additionally included in the repository are a series of prewritten Fiji macros for making *crops*, that use the Bioformats [13] plug-in to be compatible with a wide variety of imaging formats.

In this section, we describe the validation of the thresholding algorithm used by the software and cover the typical pipeline, highlighting useful features such as unbiased ROI curation. We then compare the different 2D correlation metrics available in *WormSNAP* and describe their use cases.

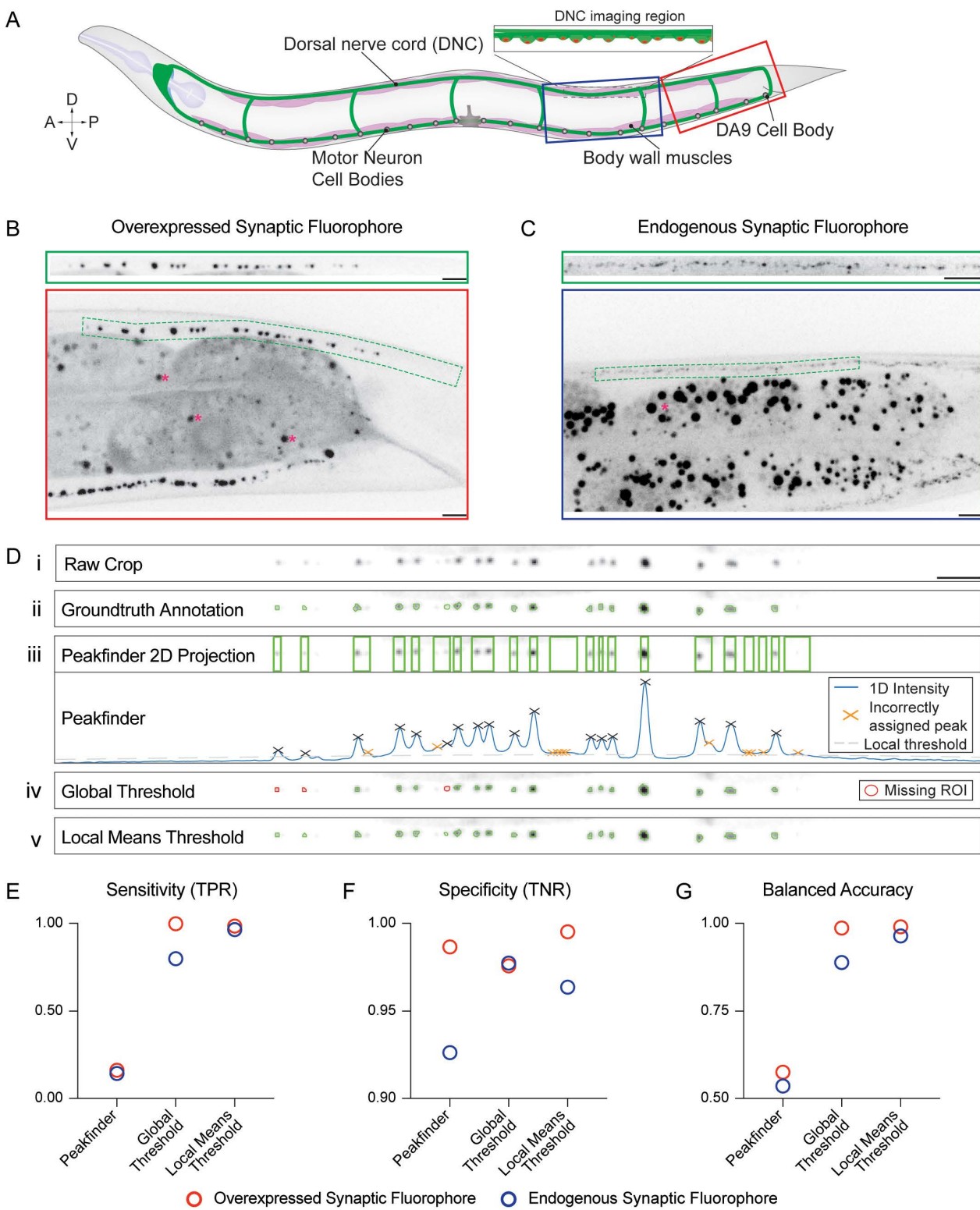

**Fig 1. Local Means Threshold shows higher accuracy than current standard ROI detection methods.** (A) Schematic of *C. elegans* body plan, with imaging regions displayed in B and C boxed in red and dark blue, respectively. (B) Top: Straightened *crop* extracted from the full-size confocal image below. Bottom: 63X confocal image of an L4-stage worm that is overexpressing the synaptic active zone marker CLA-1::GFP in the cholinergic

motor neuron DA9, whose axon extends in the dorsal nerve cord. The crop outline is shown in green. Note the nearby dark signal from the gut (dark gray), including gut granules (pink asterisks). Scale Bars, 5μm. (C) Top: Straightened *crop* extracted from the same region as displayed in the full-size confocal image below. Bottom: 63X confocal image of endogenously tagged active zone marker Neurexin::Skylan-S in the dorsal nerve cord of an L4-stage worm. Gut granules indicated with pink asterisk. Outline of crop from 100X image of the same worm shown in green. Note the lower signal-to-noise ratio compared to B. Scale Bars, 5μm. (D) Example Crop from 63X confocal image with annotated ROIs (Green) from different methods showing: *i.* The raw crop; *ii.* 2D ROIs from Ground Truth annotation; *iii.* Projected 2D ROIs from 1D peak finding algorithm, with valid peaks labeled by black Xs and erroneous peaks (not corresponding to real puncta) labeled with orange Xs; *iv.* 2D ROIs from Global Thresholding algorithm, with missing puncta high-lighted in red; and *v.* 2D ROIs from Local Means Thresholding algorithm. The local means thresholding method demonstrates the highest fidelity to the ground truth annotation. Scale Bar, 5μm. € Sensitivity (True Positive Rate) for each of the three thresholding methods, measured in the Overexpressed Synaptic Fluorophore Dataset (N = 52, red) and the Endogenous Synaptic Fluorophore dataset (N = 56, blue). (F) Specificity (True Negative Rate) for each of the same three thresholding methods, measured in the Overexpressed Synaptic Fluorophore Dataset (N = 52, red) and Endogenous Synaptic Fluorophore dataset (N = 56, blue). (G) Balanced Accuracy – the average of Sensitivity and Specificity – for each of the three thresholding methods, measured in the Overexpressed Synaptic Fluorophore Dataset (N = 52, red) and Endogenous Synaptic Fluorophore dataset (N = 56, blue).

## Thresholding and segmentation algorithm

The first step of ROI detection is thresholding, wherein regions of the image are labeled as foreground or background. Various methods exist for this purpose and can be categorized as either global thresholding, where a single threshold is used for the entire image, or adaptive thresholding, where a different threshold is computed for each pixel in an image. Adaptive thresholds have been shown to be robust to variations in background and low signal-to-noise ratios [14]. Since *crops* from *C. elegans* often exhibit changing backgrounds (due to variations in imaging depth or out-of-focus tissues), sporadic background signals from adjacent regions (e.g., autofluorescent gut granules; Fig 1B, 1C), and low signal-to-noise ratio especially for endogenous markers [15] (Fig 1C), we predicted that a local means thresholding algorithm would be most appropriate. We chose Bradley's method, in which a threshold is calculated based on the mean of pixel values in a specified neighborhood around a given pixel [16]. This thresholding algorithm can be adjusted based on the neighbor-hood size and the threshold sensitivity allowing it to be optimized for different types of fluorescent puncta.

The second step of ROI detection is segmentation, where adjacent puncta that have been grouped into a large ROI are separated into their individual ROIs. Watershed algorithms, which involve the treatment of pixel values as topography, have seen consistent use in image segmentation. The starting points for a watershed algorithm are generally based on the negative of a distance function from the edges of the ROI. We determined that an advanced watershed-based algo-rithm that uses the Fernand Meyer algorithm and considers both the distance to edge and intensity of a pixel would be most suitable to correctly separate nearby fluorescent puncta [17]. See methods for ROI segmentation formula,

## Method validation – comparison of accuracy between thresholding methods

The accuracy of the software was validated on datasets representing two extremes of fluorescent reporter type: an overexpressed transgene driving DA9 motor neuron-specific fluorescent active-zone protein expression (mig-13p:: CLA-1::GFP), which results in very bright and punctate GFP signal, and an endogenously expressed active zone local-ized fluorescent protein (NRX-1::Skylan-S [18]) which results in a dim signal. Calculating the ratio of the sum of intensities of pixels assigned (Signal) and not assigned (Noise) to ROIs of the ground-truth annotations for each dataset confirmed that the overexpressed cell-specific reporter exhibits a high signal-to-noise ratio (18.5 ± 1.9) compared to the endogenous reporter (4.09 ± 0.23).

To assess the accuracy of the software compared to the two most prevalent methods in use—1D plot profile analysis ('peakfinder') and a global thresholding method—we used receiver operating characteristics, which are based on the ratio of binary classifications of the model to that of the actual classification. These characteristics have been widely used in the assessment of a range of classification methods, such as Information Retrieval, Natural Language Processing, and Machine Learning [19,20]. In our context, they serve to visually represent the likelihood of a method to correctly classify

a true signal pixel as part of an ROI (True Positive Rate or TPR) and correctly classify a true noise pixel as background (True Negative Rate or TNR; see methods section).

We first annotated the Overexpressed fluorophore dataset using a modified version of *WormSNAP* that allows for assisted hand-drawn ROIs, creating the ground truth annotated dataset (Fig 1Dii). Each dataset was then analyzed using three different thresholding methods. In the first method, a version of *WormSNAP* using a 1D (intensity profile) based detection script was used to identify peaks, and maximal colocalization with the ground truth annotated dataset was determined by varying settings including noise-to-signal ratio, peak prominence, and maximum peak width. Since the 1D plot profile methodology was unable to assign individual pixels as ROIs/nonROIs, we also calculated the metrics in a 1-dimensional manner where a stack of pixels was assigned as ROI (or not) based on the existence of at least one ROI pixel in the stack (Fig 1Diii). The same dataset was then analyzed with a version of *WormSNAP* using an Otsu method-based global threshold [21] (Fig 1Div). Finally, the dataset was analyzed using *WormSNAP* with thresholding settings set to optimize puncta detection in the Control strain for each dataset (Fig 1Dv). This was repeated with the Endogenous fluorophore dataset (S1 Fig).

The pixel assignment by each method was compared to the hand-annotated ground truth dataset (Fig 1Dii) to calculate the following metrics: Sensitivity (True Positive Rate, or TPR), Specificity (True Negative Rate, or TNR), and Balanced Accuracy, a metric that averages the Sensitivity and Specificity to assess both aspects of the method.

We compared the *WormSNAP* local means thresholding algorithm to both the 1D peakfinder and the global threshold method in both the Overexpressed fluorophore dataset and the Endogenous fluorophore dataset (Fig 1E – 1G). Notably, we observed that while the TNR tends to be high in the 1D peakfinder, it shows significant deficits in TPR, as expected for an overly sensitive method (Fig 1E,1F). The global threshold methodology shows the opposite effect, albeit to a much smaller degree. The global threshold is very close to the local means threshold in the overexpressed dataset, demonstrating that it is adequate when dealing with high signal-to-noise images. However, in the low signal-to-noise ratio context of the endogenous reporter, the global threshold displays much lower sensitivity than the local means thresholding, as established in the literature [22]. Thus, the Balanced Accuracy metric, which combines both sensitivity and specificity, suggests that overall, the *WormSNAP* local means thresholding algorithm is superior to either the 1D peakfinder or the global threshold method across a variety of reporter types (Fig 1G).

### Method validation – Robustness test for local means thresholding

A frequent issue that arises during the development of analysis software for fluorescent imaging is over-optimization, where quantification parameters are overly constrained/fitted to show significant differences between controls and specific mutant phenotypes at the expense of identifying weaker or differing phenotypes [7]. Robustness testing can assess over-optimization by varying ROI detection and selection parameters to determine whether only certain chosen settings result in a particular conclusion.

We evaluated the robustness of WormSNAP by analyzing, for each fluorophore type (Overexpressed and Endogenous), a dataset consisting of *crops* from multiple genotypes that express the fluorophore of interest: Control worms, worms with a mutant allele resulting in a severe, easily identifiable, phenotype in the fluorescent reporter, and worms with a mutant allele resulting in a milder but still existing phenotype in the fluorescent reporter (Fig 2A,2D). The strains for each phenotype were chosen based on the extent to which they disrupted the normal distribution and/or clustering of the fluorescent puncta. In the Overexpressed fluorophore dataset (mig-13p::CLA-1::GFP), the "severe loss of puncta phenotype" mutant, *syd-1(-)*, leads to significant loss of CLA-1::GFP puncta in DA9, with the remaining puncta being much smaller and dimmer. The "mild loss of puncta phenotype" mutant, *nrx-1(-)*, leads to a loss of synapses only in the posterior region of the axon, with remaining synapses slightly smaller/dimmer. In the Endogenous fluorophore dataset, the "severe clustering phenotype" mutant, *syd-1(ΔPDZ)*, leads to significant defects in neurexin (NRX-1) clustering and thus far more diffuse NRX-1::Skylan-S puncta, whereas the "mild clustering phenotype" mutant, *syd-1(ΔC2)*, only causes a mild reduction in neurexin clustering.

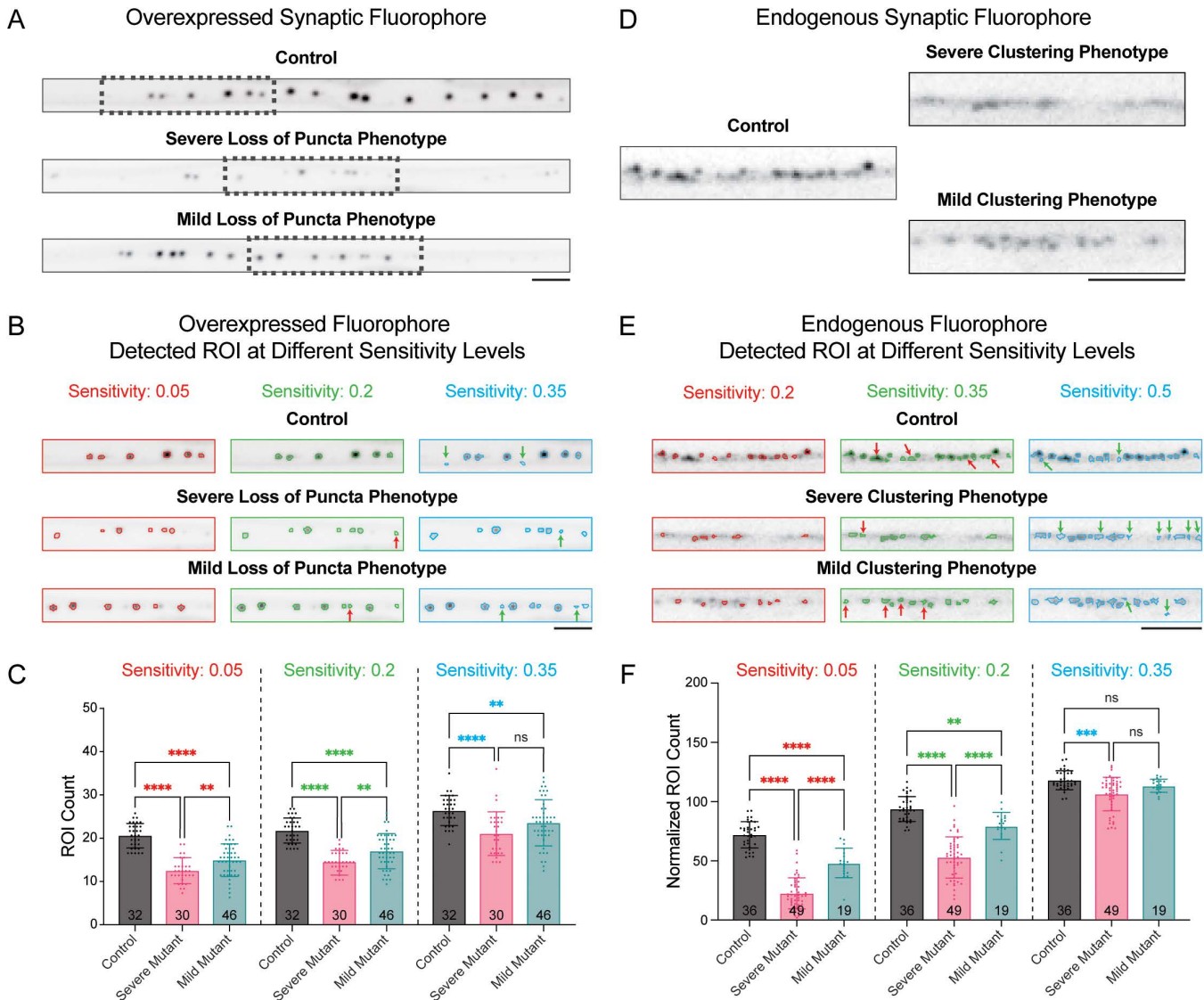

**Fig 2. Local Means Thresholding Algorithm shows high robustness.** (A) Example straightened crops of 63X confocal images from the Overexpressed Synaptic Fluorophore Dataset showing CLA-1::GFP-labeled presynapses in the DA9 neurons of control worms and worms with of two different mutant alleles that show severe [syd-1(-)] or mild [nrx-1(-)] loss of puncta. Scale Bars, 5μm. (B) Examples of ROIs detected in the boxed region of each crop in A based on minimum (0.05, red), ideal (0.2, green), and maximum (0.35, blue) sensitivity settings for ROI thresholding. Red arrows indicate ROIs that were detected at the ideal sensitivity but not at the minimum sensitivity. Green arrows indicate ROIs that were detected at the maximum sensitivity but not at the ideal sensitivity. Scale Bars, 5μm. (C) Graphs of ROI counts for the Overexpressed Synaptic Fluorophore control and mutants with loss of puncta phenotypes at the three selected sensitivities. Number of worms analyzed in each genotype shown at the bottom of each bar graph. (* = p < 0.05, ** = p < 0.01, *** = p < 0.001, **** = p < 0.0001). (D) Example sections of straightened crops of 100X confocal images from the Endogenous Synaptic Fluorophore Datasets showing NRX-1::Skylan-S-labeled presynapses in the dorsal nerve cords (DNC) of control worms and worms with of two different mutant alleles that show severe [syd-1(ΔPDZ)] or mild [syd-1(ΔC2)] disruption in NRX-1clustering. Scale Bar 5μm. (E) Examples of ROIs detected in the boxed region of the of each crop in D based on minimum (0.2, red), ideal (0.35, green), and maximum (0.5, blue) sensitivity settings for ROI thresholding. Red arrows indicate ROIs that were detected at the ideal sensitivity but not at the minimum sensitivity. Green arrows indicate ROIs that were detected at the maximum sensitivity but not at the ideal sensitivity. Scale Bars, 5μm. (F) Graphs of Normalized ROI count per 100μm (along entire crop) for Endogenous Synaptic Fluorophore control and mutants with clustering phenotypes at the three selected sensitivities. Number of worms analyzed in each genotype shown at the bottom of each bar graph. (* = p < 0.05, ** = p < 0.01, *** = p < 0.001, **** = p < 0.0001).

To test the robustness of our method, we systematically varied each of the following metrics around the original values used for quantification: local means threshold sensitivity, local means neighborhood size, minimum ROI area and maximum length-width ratio. We first optimized the settings for the control *crops* in each dataset, and then, while keeping other parameters fixed, systematically adjusted the parameter of interest around its optimal value. The sensitivity setting was varied over an interval of 0.3 (out of a total possible range of 0–1) in 0.01 increments centered around the optimal sensitivity (0.2 for the overexpressed fluorophore and 0.35 for the endogenous fluorophore; Fig 2B, 2E). The local means threshold sensitivity determines the likelihood of a given pixel to be included as signal (and thus part of ROIs) and higher values are more permissive than lower values. The ROI counts (or ROI Counts Normalized to crop length for Endogenous strains) of each genotype in the dataset was recorded, as well as the p-value derived from a one way Anova. We then plotted bar graphs for the minimum, ideal, and maximum sensitivity considered (Fig 2C, 2F).

The robustness testing showed that the settings used were not overly sensitive to the comparisons made in this paper for both the overexpressed synaptic fluorophore (Fig 2B, 2C) and the endogenous synaptic fluorophore (Fig 2E, 2F). While the individual output values differed notably as the sensitivity was changed, the conclusion of whether the severe mutant genotypes were significantly different from the control remained unchanged (Fig 2C, 2F). In fact, we only saw noteworthy increases in p-value for the Endogenous severe phenotype mutant (Fig 2F). However, as expected from mutants that show phenotypes closer to the control genotype, there was a significant increase in p-value as sensitivity was increased for the mild phenotype mutants, with the effect being clearer in the Endogenous mild phenotype mutant which cannot be distinguished from the control at the most permissive sensitivity level (Fig 2F). Notably, we can see that we can distinguish the mild mutants from the severe mutants at the optimal and lower sensitivity levels. In images of example *crops* overlaid with ROIs using different parameter settings (highlighted in red, green and blue in Fig 2B, 2E), it appears that the higher sensitivity settings falsely include noise as puncta (green arrows), especially in the severe mutant, thereby inflating the mutant ROI count leading to the inability to distinguish the mutants from each other and the control. On the other hand, while puncta are lost (red arrows) when using lower sensitivity settings, they tend to be lost rather uniformly, which does not impact the ability to distinguish between phenotypes as much.

## WormSNAP graphics user interface

**GUI pipeline.** Upon startup, *WormSNAP* prompts users to select a folder containing preprocessed *crops*. The *WormSNAP* GUI consists of two sections: a display panel and an input panel. The display panel shows *crops* ordered by genotype and number, while the input panel serves as the primary way to interact with the GUI. The input panel contains three tabs that users proceed through sequentially. First, the Preprocessing tab (S3A Fig) allows users to name the image channels, rename genotypes, and, if necessary, specify the magnification/resolution.

Users then proceed to the ROI detection tab (S3B Fig) to specify the thresholding and restriction settings for each channel. The restriction parameters help exclude spurious ROIs that arise from imaging artifacts or noise (Fig 3A). The first set of restriction parameters, called Crop Restriction Parameters, enable the exclusion of ROIs based on their proximity to the edges of *crops*, which helps exclude artifacts such as *C. elegans* autofluorescent gut granules (Fig 3B). The second set of parameters filters ROIs based on features including area, circularity, length-to-width ratio, and intra-ROI intensity variance. These criteria allow users to exclude noise that often appears as a few bright pixels (Fig 3C).

After ROIs are detected in all available channels, users proceed to the Results tab (S3C Fig) to visualize the ROIs and configure output settings. Using the ROI Display panel, users can easily view their crops and overlaid ROIs (Fig 3D – 3G). When viewing 2-channel images, users can view ROIs of each channel (Fig 3E,3G) as well as mix and match between ROIs and Channels (Fig 3H), and even view ROIs from both channels overlaid on top of a crop (Fig 3I).

*WormSNAP's* output options (S3C Fig) include.csv files of selected quantification parameters, a combined.csv file containing ROI-specific data for further analysis, a '.mat' file that allows you to easily replicate previous analyses, and PDF-ready files of montages or individual *crops* with ROI overlays of the user's choice (S3D Fig). A step-by-step walkthrough of the software is available in the GitHub documentation.

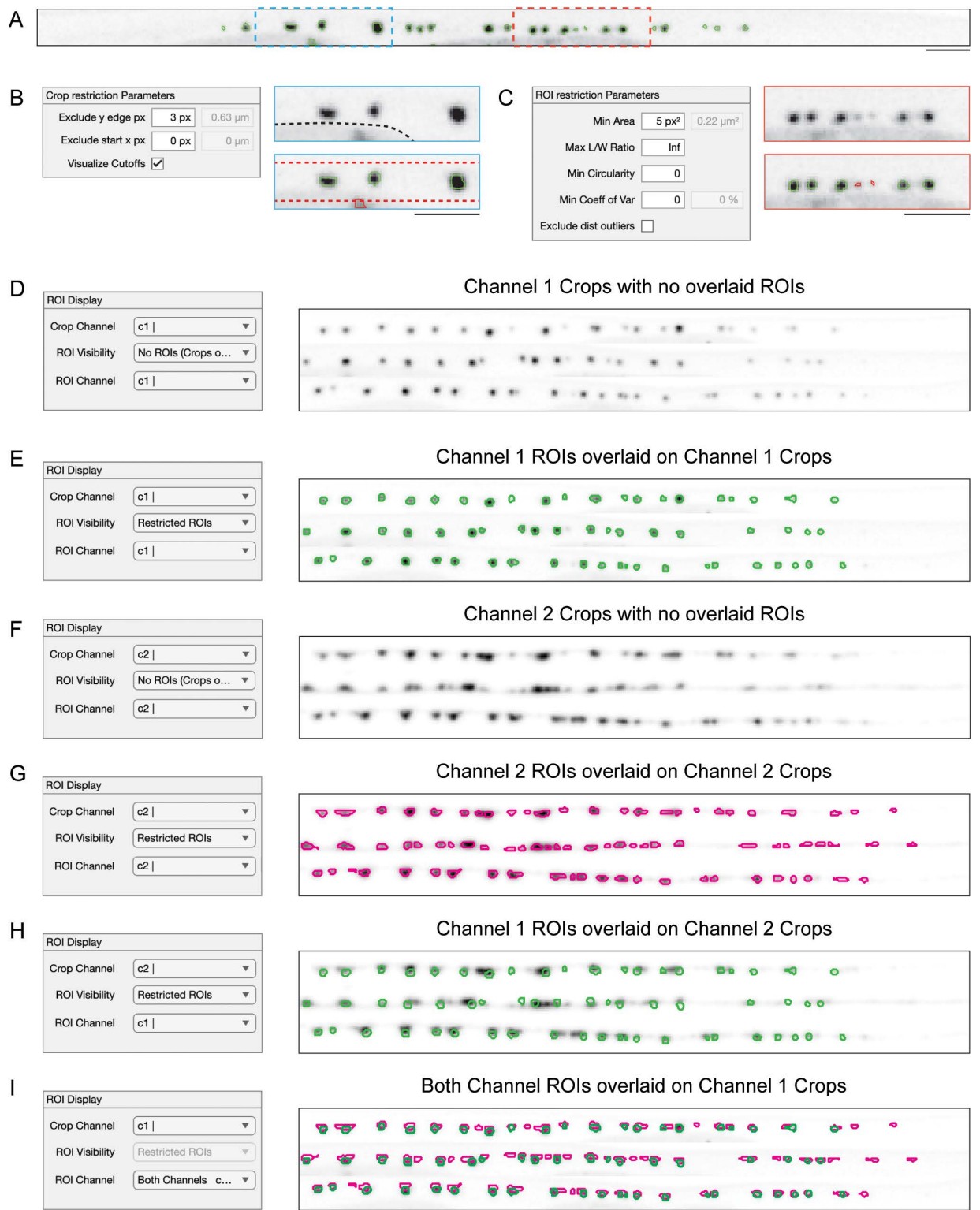

**Fig 3. ROI Restriction and Visualization in WormSNAP GUI.** (A) ROIs detected by Local Area Thresholding algorithm on the crop of an Overex-pressed Synaptic Fluorophore (previously shown in Fig 1B). (B) Left:Crop Restriction parameters from WormSNAP's ROI Detection Tab (see S3B Fig), with a y-edge exclusion of 3px (≈0.63μm). Right Top: Magnified view of the blue box (in A) showing no ROIs assigned and illustrating that the

bottom-most punctum is part of the gut (outlined in dotted black line). See Fig 1B for the full image showing the gut impinging on the crop outline. Right Bottom: Example of how excluding y-edge pixels can remove a misassigned ROI from the gut granule. Dotted lines indicate the boundaries beyond which ROIs are ignored. The excluded ROI is shown in red. Scale Bar, 5µm. (C) Left: ROI Restriction parameters from WormSNAP's ROI Detection Tab, set to a minimum area of 5px$^2$/0.22µm$^2$. Right Top: Magnified view of the red box (in A), showing no ROIs assigned. Right Bottom: Example of how the minimum area setting excludes misassigned ROIs that appear as individual bright pixels due to noise. Excluded ROIs are shown in red. Scale Bar, 5µm. (D-I) Magnified view of WormSNAP's Results Tab (see S3C Fig) and an example 2-channel crops visualization of 3 crops from the Display Tab, showing: D. Channel 1 (CLA-1::GFP) intensities; E. Channel 1 (CLA-1::GFP) intensities overlaid with ROIs from Channel 1 (CLA-1::GFP) in green; F. Channel 2 (RAB-3::tdTomato) intensities; G. Channel 2 (RAB-3::tdTomato) intensities overlaid with ROIs from Channel 2 (RAB-3::tdTomato) in magenta; H. Channel 2 (RAB-3::tdTomato) intensities overlaid with ROIs from Channel 1 (CLA-1::GFP) in green; I. Channel 1 (CLA-1::GFP) intensities overlaid with Channel 1 (CLA-1::GFP) ROIs in green and Channel 2 (RAB-3::tdTomato) ROIs in magenta. Scale Bars, 5µm.

**GUI validation – scalability.** Scalability of a method is important for widespread adoption. We optimized the *WormSNAP* GUI to analyze datasets containing more than 100 *crops*. One of the GUI's main advantages is the ability to easily view many *crops* at once; however, this can be computationally intensive, leading to software slowdowns or crashes for large datasets. To address this, we implemented a tab system to limit the maximum number of *crops* displayed simultaneously. We compared a single-tab display of all *crops* to a split-tab display (with 50 crops per tab) for datasets of 50–200 *crops* (S3E, S3F Fig). Both the time-to-display and ROI calculation time decreased when using split tabs instead of a single tab as the total number of *crops* increased, demonstrating that the split tab method is more effective for displaying large datasets.

## Colocalization metrics

In fluorescence microscopy, assessing colocalization between two fluorescent signals is crucial for understanding biological interactions of the underlying proteins [23]. Hence, WormSNAP also calculates correlation metrics for 2D images as part of its output (Fig 3C,3D). Two metrics widely used in the field are the Pearson Correlation Coefficient (PCC) and the Manders' Coefficients, M1 and M2 [24].

The PCC quantifies the linear relationship between the intensities of two channels, ranging from -1–1, where values close to 1 indicate a strong positive correlation [25]. It accounts for the pixel intensities of each channel over the entire image. In unprocessed images, the PCC can be severely impacted by ordered background fluorescence - such as that caused by the sample's 3D geometry - as well as by noise [26]. However, because our workflow preprocesses images into *crops* that minimize background area, we can substantially reduce the impact of noise and unrelated structured background (S4A – S4C Fig). For our crop images, the main limitation of PCC lies in its single-metric output encompassing both channels, rendering it incapable of distinguishing which channel exhibits disruption.

In contrast, Manders' Coefficients measure the fraction of signal in one channel that overlaps with the signal in the other, with M1 quantifying the fraction of signal pixels in channel 1 that are also signal pixels in channel 2, and M2 measuring overlap in the opposite direction [27]. Because M1 and M2 provide channel-specific information, they are well suited to robust thresholding algorithms such as the local means algorithm used by *WormSNAP*.

*WormSNAP* also calculates a novel third set of metrics, the ROI Overlap Ratios, R1 and R2. R1 is the fraction of ROIs (as opposed to pixels) in channel 1 that overlap with ROIs in channel 2, above a user-defined Overlap Threshold. The Overlap Threshold defines the minimum pixel overlap needed to classify two ROIs as overlapping; a lower threshold makes partially overlapping markers more likely to be considered co-localized. R2 measures the overlap in the opposite direction.

To assess changes in the colocalization pattern between two proteins that occupy different regions *within* the same structure, such as active zone proteins (e.g., CLA-1) localized to a distinct region within the synaptic bouton, and synaptic vesicle markers (e.g., RAB-3) that fill the bouton, the Manders' Coefficients are more suitable. This is because they quantify the *relative* overlap between ROIs (Fig 4A), allowing researchers to determine whether this balance is altered

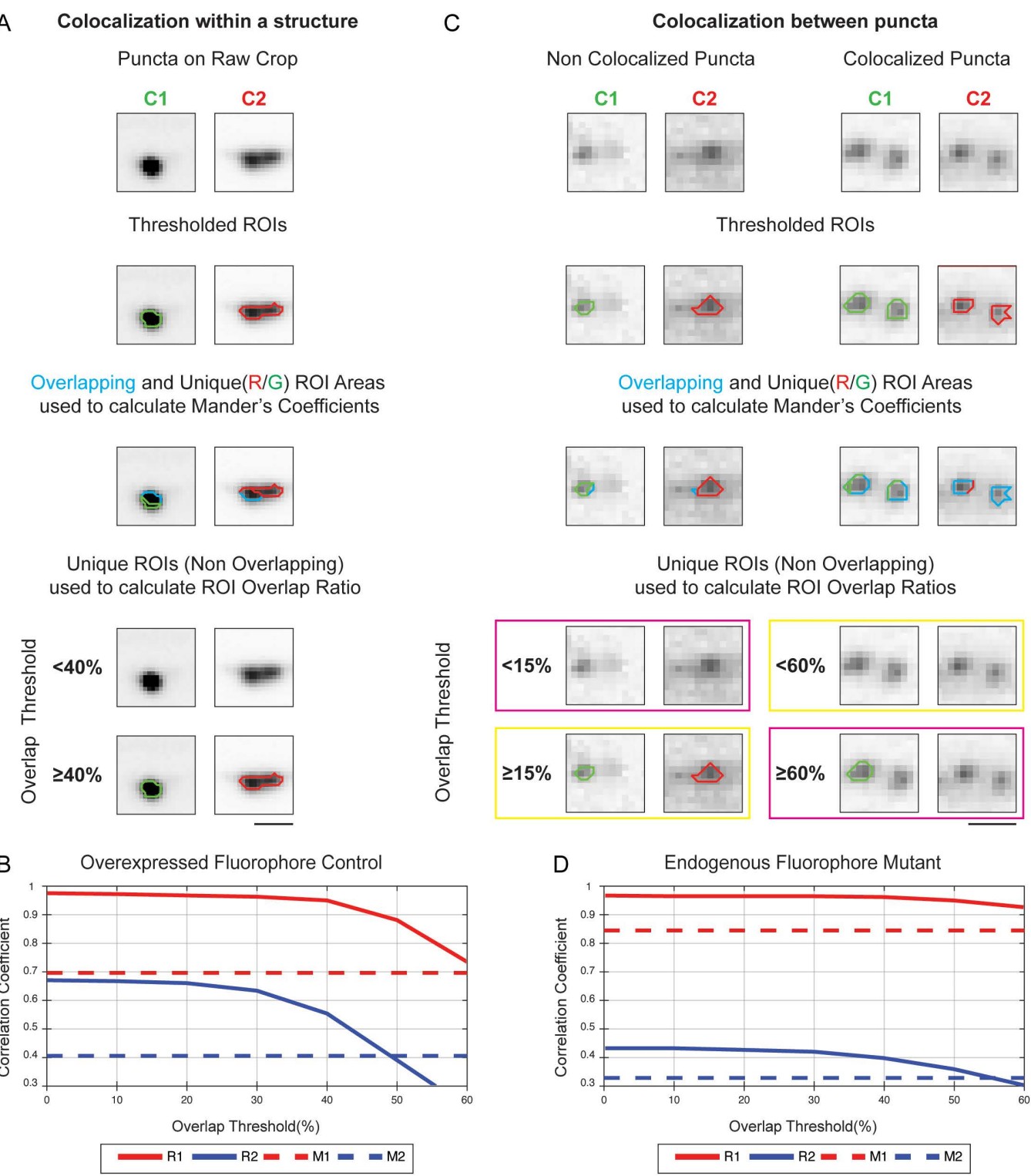

**Fig 4. Comparison of WormSNAP Correlation Metrics.** (A) Example of two proteins (CLA-1, an active zone marker, and RAB-3, a synaptic vesicle marker) that exhibit different localization patterns *within* the same synaptic structure. The Mander's coefficient is calculated based on the ratio of overlap (shown in blue) relative to the total area of the thresholded ROI. Mander's coefficients accurately capture these differences because the extent of overlap between the channels is markedly different. In contrast, the ROI Overlap Ratio, which provides binary information for individual puncta, either shows

complete colocalization at Overlap Thresholds below 40%, or none at Overlap Thresholds of 40% or more. Therefore, for assessment of colocalization *within* puncta (or structures), Mander's coefficient are more suitable. Scale Bar, 2µm. (B) Graph of ROI Overlap Ratio (R1 and R2) for the Overexpressed fluorophore Control worms shown in A which express fluorescent CLA-1 and RAB-3 plotted against Overlap Threshold showing that both R1 and R2 are much higher than the respective Mander's Coefficients (M1 and M2), with a strong decrease at 40% in both. This demonstrates on a macroscopic scale that Mander's coefficients are an easier way to evaluate localization patterns *within* a structure. (C) Example of two proteins (CLA-1 and NRX-1) that exhibit different localization patterns across synapses in a mutant background – i.e., they are no longer found at the same synapses. Two scenarios are shown: non-overlapping puncta (left) and overlapping puncta (right). Mander's Coefficients yield a non-zero value for the non-overlapping puncta, owing to minor boundary overlap, and a value below 1 (i.e., less than complete colocalization) for the overlapping puncta. In contrast, the ROI Overlap Metrics use an Overlap Threshold to determine colocalization. Correct assignments by the ROI Overlap Metrics are indicated by the yellow rectangle; incorrect assignments are pink. Non-overlapping puncta are correctly identified at Overlap Thresholds of 15% or higher, while overlapping puncta are correctly identified below an Overlap Threshold of 60% - indicating that the optimal Overlap Threshold for this dataset lies between 15% to 60%. Scale Bar, 1µm. (D) Graph of ROI Overlap Ratio (R1 and R2) for the entire dataset of Endogenous fluorophore Mutant worms shown in C that exhibit different localization patterns of NRX-1 and CLA-1 across synapses plotted against Overlap Threshold. The graph shows that in the mutant, the inflection points are not identical, allowing us to differentiate between colocalized and non-colocalized puncta with more granularity than M1 and M2.

in a particular mutant. This is evident when looking at the genotype wide correlation coefficient for the Overexpressed Reporter (Fig 4B), where we see that R1 and R2 values change in tandem, not giving us any more information than the Manders' Coefficients at any given Overlap Threshold.

To determine whether two proteins are located within the same structure – for example, whether different synapses contain one protein versus another – the ROI Overlap Ratios are preferred. This is because they assign individual, binary colocalization values for individual ROIs and disregard small overlaps between adjacent but non-overlapping ROIs in different channels (Fig 4C). The ROI Overlap Ratios can be further tuned for specific pairs of markers by setting an Overlap Threshold that distinguishes non-overlapping puncta as separate while still classifying truly overlapping puncta together (in Fig 4C, this Overlap Threshold ranged from 15% - 60%). The genotype wide correlation coefficients (Fig 4D), demonstrates this fine tuning with R1 and R2 showing different inflection points, which would allow us to separate out colocalized versus non colocalized puncta.

Noise is a major consideration when selecting a correlation metric [23]. We generated synthetic two-channel datasets based on Channel 1(CLA-1::GFP) of the Overexpressed Synaptic Fluorophore control *crops* with four different noise types designed to mimic typical confocal microscopy issues: Shot Noise (Simulated using a Poisson function), Thermal noise (simulated with Gaussian functions), Salt and Pepper/Impulse noise, and Speckle/Multiplicative noise (S4D Fig). The PCC, M1, and R1 were then computed to measure the correlation between the original image and the noisy image. Because the PCC ranges from -1–1 – twice the range of M1 and R1 (which span [0,1]) – it was normalized to [0,1]. Next, the correlation metrics were plotted for the different noises, with the parameters used the generate the noise varied when appropriate, to determine which metric remained closest to perfect correlation (=1) under various noisy conditions. PCC showed the least deviation overall but was heavily impacted by salt and pepper noise. R1 outperformed M1 in all noise types except for Poisson where they were comparable (S4E – S4H Fig).

In summary, *WormSNAP* provides a range of colocalization metrics that balance accuracy and noise sensitivity, enabling selection of the most suitable measurement for each dataset and scientific question.

## Discussion

In this paper, we introduce *WormSNAP*, an open-source graphical user interface (GUI) for no-code, local means-based region of interest (ROI) detection, optimized for analysis of large datasets of fluorescent puncta in *C. elegans*. We highlight the strong performance of our local means thresholding algorithm relative to existing methods, especially across a range of fluorescent markers with varying signal-to-noise ratios. To our knowledge, this is the first GUI that allows simultaneous display and interaction with images and 2D ROIs from entire datasets. We have also incorporated multiple user-friendly features such as saving thresholding parameters, renaming channels, managing outliers, selecting correlation

metrics, exporting data for easy plotting, and saving images in figure-friendly formats. Through these advancements, we aim to standardize puncta detection in *C. elegans* research and substantially shorten the time from image acquisition to publishable data, ultimately accelerating research workflows.

### Interpretation of results

By comparing model accuracy with other methods, we demonstrate that WormSNAP's local means thresholding algorithm is well-suited for puncta detection and characterization when compared to the most used methods. In addition, our results show that WormSNAP's underlying methodology is highly robust: user-modifiable settings support the quantification of a wide range of fluorophores imaged in *C. elegans* nerve cords and more complex synapses.

### Advantages and limitations of the software

Our software offers direct control over ROI detection and restriction parameters without coding requirements, as well as the ability to analyze large datasets. Thus, users can easily optimize the puncta detection settings while avoiding the barriers of specialized programming knowledge. Additionally, our methodology exhibits consistent improvement over widely used approaches, such as 1D plot profiles and global threshold-based ROI detection. With various ease-of-use features, users can annotate datasets and obtain useful outputs without recalculating data or learning how to process esoteric file types.

Currently, WormSNAP is limited to 2D puncta detection of up to 2 channels at a time, constraining its applicability to certain multicolor imaging strategies [28]. Also, the preprocessing step of drawing *crops* precludes a fully automated pipeline. However, time spent analyzing images is still significantly reduced, because *crops* need to be drawn only once per dataset. Analysis parameters and exclusions are also saved after the session is done, allowing for re-analysis of older data using new optimized parameters.

Although other software (e.g., *WormPsyQi* [29]) can detect puncta in 3D, it requires overexpressed cytoplasmic markers to define the axon boundary, and the synapse detection model may require further training to function. In contrast, *WormSNAP* was designed as an easy-to-use software that can be applied to existing fluorescent imaging datasets without axonal boundary demarcation. Moreover, *WormSNAP* lets users display large datasets effortlessly and repeat the same analyses with a single click.

### Future directions and potential improvements

Future work will focus on extending our approach to 3D images. Currently, maximum intensity projection from 3D image stacks is used, but for *C. elegans* nerve cords (~40 nm thick), the need for 3D data is relatively low [30]. Ongoing efforts aim to enable 3D thresholding in future releases. Additional improvements under development include blinding functions and automated puncta detection in standard confocal images using feature detection methods.

## Methods

### *C. elegans* methods

*C. elegans* strains were derived from the Bristol strain N2 and raised at 23°C on NGM plates seeded with OP50 *E. coli*, according to standard protocols [31].

The 'Overexpressed Cell Specific Fluorescent Protein' (Overexpressed fluorophore) datasets were collected by imaging CLA-1::GFP (a presynaptic active zone marker [4]) in the axon of the DA9 neuron from the following strains: TV18675 [*wyIs685 V*], a strain containing an integrated transgene that expresses CLA-1::GFP driven by the DA9 neuron-specific mig-13 promoter; TV22469 [*wyIs685, nrx-1(wy1155) V*] ('Mild Loss of Puncta Phenotype'), which expresses the same transgene in a *nrx-1(-)* background showing loss of posterior axonal CLA-1::GFP punta [12]; and PTK323 [*wyIs685 V;*

*syd-1(ju82) II*]¹² ('Severe Loss of Puncta Phenotype'), which expresses the same transgene in a *syd-1(-)* background, leading to loss of most CLA-1::GFP puncta.

The 'Endogenous Fluorescent Protein' (Endogenous fluorophore)dataset was collected by imaging endogenously-tagged NRX-1::Skylan-S in the dorsal nerve cord of worms from the following strains: PTK196 [*syd-1(kur33) II; nrx-1(ox719) V*] ('Control'), which expresses endogenously tagged NRX-1::Skylan-S and endogenously tagged SYD-1::mScarlet; PTK362 [*syd-1(kur75) II; nrx-1(ox719) V*] ('Mild Clustering Phenotype'), which expresses endogenously tagged NRX-1::Skylan-S and endogenously tagged SYD-1(ΔC2)::mScarlet and exhibits reduced number and intensity of NRX-1::Skylan-S puncta; and strain PTK270 [*syd-1(kur57) II; nrx-1(ox719)V*] ('Severe Clustering Phenotype'), which expresses endogenously tagged NRX-1::Skylan-S and endogenously tagged SYD-1(ΔPDZ)::mScarlet and exhibits a severe disruption of NRX-1::Skylan-S clustering.

The dataset used in Fig 4B and 4D was collected by imaging NRX-1::Skylan-S and CLA-1::mScarlet in the dorsal nerve cord of worms from strain PTK501 [*cla-1(kur27) IV; nrx-1(kur64) V*], which expresses endogenously tagged CLA-1::mScarlet and Myristoylated::NRX-1(Intracellular Domain)::Skylan-S in the NRX-1 locus and exhibits loss of synaptic (CLA-1 colocalized) NRX-1 puncta.

Images of non-nerve cord synaptic puncta were collected from PTK447 [*syd-1(wy1320) II; wyIs891 III; ppk-1(ox874); oxSi1275 IV*], which expresses SYD-1::FLPon-mScarlet in the HSN neuron with the HSN-specific unc-86 promoter driving FLP recombinase.

## Data acquisition and crop tracing

Confocal images of fluorescently labeled proteins were collected at room temperature in living, anesthetized *C. elegans.* L4.4-stage [32] hermaphrodites were anesthetized in 20 mM levamisole (Sigma-Aldrich) in M9 buffer, mounted on 10% agarose pads, and sealed under a #1.5 coverslip with petroleum jelly.

Images were acquired on a spinning disk confocal (3i) equipped with an Evolve 512 EMCCD camera (Photometrics) and a CSU-X1 M1 spinning disk (Yokogawa) mounted on an inverted Zeiss Axio Observer Z1 microscope. The 'Overexpressed Cell Specific Fluorescent Protein' datasets were collected with a Zeiss Plan-Apochromat 63x/1.4NA oil immersion objective in z-stacks of the DA9 axon in the dorsal nerve cord, with 0.27 μm z steps over a 5 μm z range. The 'Endogenous Fluorescent Protein' images were collected with a Zeiss Plan-Apochromat 100x/1.4NA oil immersion objective in z-stacks from the dorsal nerve cord, with 0.27 μm z steps over a 6 μm z range. Z-stacks were obtained centered around the highest in-frame fluorescence of either the DA9 axon or DNC using non-saturating laser, exposure and camera settings. Synapses in the HSN neuron were collected in z-stacks with 0.27 μm z-steps over an 8.4 μm z range, also with a 100x/1.4NA oil immersion objective. Images were acquired using Slidebook 6.0 software (3i).

During initial image processing procedures, raw z-stack volumes from the native.sld Slidebook file format (3i) were rendered into 16-bit maximum intensity projections tifs, and 20-pixel-wide lines were drawn over synaptic regions of interest and straightened into 32-bit tif cropped images using Fiji macro 'Batch Axon Tracer - sld' included in the Github repository.

## ROI segmentation

The formula we used to get the Segmented ROIs is as follows:

$$SegmentedROIMask = Watershed(-((bwdist(\sim ROIMask)).*normImg)$$

The distance transform of the inverse of the binary image produced by the initial ROI Mask is calculated to obtain the distance information of each ROI pixel relative to the non-ROI pixels. Then, the dot product of the distance transform is multiplied with the normalized image (normImg), adding in information about the relative pixel intensity in each ROI pixel. The watershed function is then applied, producing ROIs segmented based on both on the location and intensity of the ROI pixels.

## Method validation – Robustness test for local means thresholding

Using a custom ROI overlap analysis script, the different methods were compared *crop*-wise to the ground truth annotated dataset, by evaluating three standard receiver operating characteristics for accuracy comparisons, namely the Sensitivity/True Positive Rate (TPR), Specificity/True Negative Rate (TNR) and Balanced Accuracy (BA).

The receiver operating characteristics were calculated as follows:
Sensitivity/True Positive Rate (TPR) was assessed as:

$$True\ Positive\ Rate\ (TPR)\ =\ \frac{True\ ROI\ px}{Total\ number\ of\ ROI\ px}$$

where 'True ROI px' refers to an ROI pixel that colocalized with a ground-truth annotated ROI pixel,
Specificity/True Negative Rate (TNR) was assessed as:

$$True\ Negative\ Rate\ (TNR) = \frac{True\ nonROI\ px}{Total\ number\ of\ nonROI\ px}$$

where 'True nonROI px' refers to a pixel that was not considered as part of an ROI by either the software or the ground-truth annotation.

Balanced Accuracy (BA) was assessed as follows:

$$Balanced\ Accuracy\ (BA) = \frac{TPR + TNR}{2}$$

## 2D channel correlation analysis – correlation metrics

*WormSNAP* calculates three different correlation metrics: the Pearson's Correlation Coefficient, the Manders' Coefficients and the ROI Overlap Ratios.

The Pearson's Correlation Coefficient is calculated directly from ROI intensity as follows for a two-channel image:

$$PCC = \frac{\sum_{i=1}^{N} (C1_i - \mu_{C1})(C2_i - \mu_{C2})}{\sqrt{\sum_{i=1}^{N}(C1_i - \mu_{C1})^2 \sum_{i=1}^{N}(C2_i - \mu_{C2})^2}}$$

Where:
   N is the total number of pixels in the image
   $C1_i$ is the intensity of pixel *i* in Channel 1
   $\mu_{C1}$ is the mean intensity in Channel 1
   $C2_i$ is the intensity of pixel *i* in Channel 2
   $\mu_{C2}$ is the mean intensity in Channel 2

The Manders' Coefficients and ROI Overlap Ratios are calculated based on the ROI assignments post thresholding and ROI restriction.

The Mander's Coefficients (M1 and M2) are calculated as follows for a two-channel image [27]:

$$M1 = \frac{\sum_{i=1}^{N} C1_{i,coloc}}{\sum_{i=1}^{N} C1_i}$$

$$M2 = \frac{\sum_{i=1}^{N} C2_{i,coloc}}{\sum_{i=1}^{N} C2_i}$$

Where:

N is the total number of pixels in the image

$C1_i = 1$ if pixel $i$ is thresholded as signal in Channel 1 and $C1_i = 0$ otherwise

$C2_i = 1$ if pixel $i$ is thresholded as signal in Channel 1 and $C2_i = 0$ otherwise

$C1_{i,coloc} = 1$ if pixel $i$ is thresholded as signal in both channels and $C1_{i,coloc} = 0$ otherwise

$C2_{i,coloc} = 1$ if pixel $i$ is thresholded as signal in both channels and $C2_{i,coloc} = 0$ otherwise

The ROI Overlap Ratios (R1 and R2) are calculated as follows for a two-channel image:

$$R1 = \frac{\sum_a C1_{a,coloc}}{N1}$$

$$R2 = \frac{\sum_b C2_{b,coloc}}{N2}$$

Where:

N1 is the number of ROIs detected in channel 1

N2 is the number of ROIs detected in channel 1

$C1_{a,coloc} = 1$ if ROI $a$ is considered to have significant overlap with ROIs in Channel 2 based on an Overlap Threshold of the percentage of its pixels thresholded as signal in Channel 2, $C1_{a,coloc} = 0$ otherwise

$C2_{b,coloc} = 1$ if ROI $b$ is considered to have significant overlap with ROIs in Channel 1 based on an Overlap Threshold of the percentage of its pixels thresholded as signal in Channel 1, $C2_{b,coloc} = 0$ otherwise

### 2D channel correlation analysis – synthetic noise

Noise was added to original images using MATLAB's Image Processing Toolbox's *imnoise* function [33]. The function has inbuilt settings for salt and pepper noise and speckle noise while thermal noise was modeled using a Gaussian distribution and shot noise was modeled using a Poisson distribution [33]. The applied Gaussian noise was varied both in terms of mean (0 to 0.2 keeping variance at 0.01) and variance (0 to 0.2, keeping mean at 0). The variance of the random uniformly distributed random function used speckle noise was varied from 0 to 0.2 and the noise density of the Salt and Pepper Noise was varied from 0 to 0.05.

### Statistical analysis

All statistical analysis was done using GraphPad Prism version 10.5.0 for MacOS, GraphPad Software, www.graphpad.com. For Fig 2, Brown-Forsythe and Welch One Way ANOVA tests followed by Dunnett's T3 multiple comparisons test was done and the resulting p-value plotted as appropriate. Analysis for S2B and S4C Figs was done using an unpaired Welch's t-test. For S4A Fig, a paired two-tailed Student's *t test* was used to calculate the p-values.

### Supporting information

**S1 Fig. ROI Thresholding Methods in Endogenous fluorophore images.** Example Crop from 100X confocal images of endogenous synaptic marker in the dorsal nerve cord (Neurexin::Skylan-S) with annotated ROIs (green) from different methods showing: *i*. The raw crop with a region highlighted; *ii*. A zoom in of the raw crop in the highlighted region; *iii*. 2D

ROIs from the Ground Truth annotation; *iv.* Projected 2D ROIs from 1D peak finding algorithm, with valid peaks labeled by black Xs and unlabeled puncta labeled with red Xs; *v.* 2D ROIs from Global Thresholding algorithm, with erroneous puncta in orange and missing puncta highlighted in red; and *vi.* 2D ROIs from Local Means Thresholding algorithm, also showing missing puncta highlighted in red. The local means thresholding method demonstrates the highest fidelity to the ground truth annotation. Scale Bars shown on *i* and *ii*, 5µm.
(TIF)

**S2 Fig. Local Means Threshold in nonlinear axons.** (A) Example *crops* of HSN synapses from worms with endogenous expression of a synaptic fluorophore (SYD-1::mScarlet). Top: *Crops*. Bottom: *Crops* with ROI overlay (green). Scale, 3 µm. (B) Quantification of number of SYD-1::mScarlet ROIs (normalized to 100µm$^2$) in HSN synapses from two different imaging slides. The lack of significant difference between the two samples demonstrates replicability of local means thresholding in nonlinear axons.
(TIF)

**S3 Fig. WormSNAP GUI Display and Outputs.** (A) Screenshot of the Preprocessing Tab in the Graphic User Interface (GUI). (B) Screenshot of the ROI Detection Tab in the GUI. (C) Screenshot of the Results Tab in the GUI. (D) Example Output folder for a 2-channel dataset generated using the output settings shown in (C). (E) *Crop* Display Speed, measured in seconds per *crop,* for datasets of various sizes displayed in a single-tab versus a split-tab. The split-tab approach limits each tab to a maximum of 50. (F) ROI Calculation speed (seconds per *crop)* for datasets of various sizes displayed in a single-tab versus a split-tab.
(TIF)

**S4 Fig. Effects of cropping and noise on WormSNAP Correlation Metrics.** (A) Example 2 channel confocal image from a control worm from the endogenous fluorophore dataset with nrx-1::Skylan-S in Channel 1 and syd-1::mScarlet in channel 2 with the corresponding 2-channel *crop*. Scale Bars, 5µm. (B) Pearson's Correlation Coefficient (PCC) for original images and resulting *crops* for the control worms in the endogenous fluorophore dataset (N = 33) (* = $p < 0.05$, ** = $p < 0.01$, *** = $p < 0.001$, **** = $p < 0.0001$). (C) PCC for original images and resulting *crops* of the control and severe clustering mutant (*syd-1(ΔPDZ))* from the Endogenous dataset, demonstrating that cropping images improves the PCC metric for control but not for the severe clustering mutant phenotype. Consequently, *crops* show a significant difference in PCC between the two strains, whereas original images do not (* = $p < 0.05$, ** = $p < 0.01$, *** = $p < 0.001$, **** = $p < 0.0001$). (D) Image of a *crop* from the Overexpressed Synaptic Fluorophore Dataset showing Channel 1 (CLA-1::GFP) and the same crop after different types of noise was applied to the signal. Poisson noise was generated using a Poisson function with mean equivalent to signal pixel intensity; Gaussian noise was generated using a gaussian function with mean 0 and variance 0.01; speckle noise was generated using a uniformly random noise function with mean 0 and variance 0.05; salt and pepper noise was generated using a noise density of 0.05 (5% of pixels replaced). illustrating how different noise types affect signal in *crops*. Scale Bar, 5µm. (E) Quantification of three Correlation Metrics – PCC (normalized to [0,1] from its original range of [-1,1), Manders' Coefficient (M1), and ROI Overlap ratio (R1) for synthetic 2-channel images consisting of the original channel 1 of the Overexpressed Synaptic Fluorophore Dataset (N = 52) and channel 2 generated by applying Poisson noise (used to simulate shot noise) to the first channel. The mean and standard deviation of each metric was plotted. (F) Quantification of the three Correlation Metrics in D for Gaussian Noise (used to simulate thermal noise). (Top) Means and standard deviations of the Correlation Metrics plotted against mean of added gaussian noise (variance = 0.01). (Bottom) Means and standard deviations of the Correlation Metrics plotted against variance of added gaussian noise (mean = 0). (G) Means and standard deviations of the three Correlation Metrics in D for speckle noise plotted against variance of the uniformly random noise function (mean = 0) used to simulate the noise. (H) Means and standard deviations of the three Correlation Metrics in D for salt and pepper noise plotted against noise density used to simulate the noise.
(TIF)

**S1 Table. Strain List.**
(DOCX)

## Author contributions

**Conceptualization:** Araven Tiroumalechetty, Peri T. Kurshan.

**Data curation:** Araven Tiroumalechetty.

**Formal analysis:** Araven Tiroumalechetty.

**Investigation:** Araven Tiroumalechetty, Elisa B. Frankel.

**Methodology:** Araven Tiroumalechetty.

**Project administration:** Peri T. Kurshan.

**Resources:** Peri T. Kurshan.

**Software:** Araven Tiroumalechetty.

**Supervision:** Peri T. Kurshan.

**Validation:** Araven Tiroumalechetty, Elisa B. Frankel.

**Visualization:** Araven Tiroumalechetty, Elisa B. Frankel.

**Writing – original draft:** Araven Tiroumalechetty.

**Writing – review & editing:** Elisa B. Frankel, Peri T. Kurshan.

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
