## [Decision Letter · Decision Letter 0]

9 Jul 2025

PCOMPBIOL-D-25-00888

WormSNAP: A software for fast, accurate, and unbiased detection of fluorescent puncta in C. elegans

PLOS Computational Biology

Dear Dr. Kurshan,

Thank you for submitting your manuscript to PLOS Computational Biology. After careful consideration, we feel that it has merit but does not fully meet PLOS Computational Biology's publication criteria as it currently stands. Therefore, we invite you to submit a revised version of the manuscript that addresses the points raised during the review process.

Please submit your revised manuscript within 30 days Sep 08 2025 11:59PM. If you will need more time than this to complete your revisions, please reply to this message or contact the journal office at ploscompbiol@plos.org. Please include the following items when submitting your revised manuscript:

We look forward to receiving your revised manuscript.

Kind regards,

Adriana San Miguel

Academic Editor

PLOS Computational Biology

Feilim Mac Gabhann

Editor-in-Chief

PLOS Computational Biology

**Additional Editor Comments (if provided):**

**Journal Requirements:**

**Reviewers' comments:**

Reviewer's Responses to Questions

**Comments to the Authors:**

Reviewer #1: In this manuscript, Tiroumalechetty and colleagues introduce a new software package for the quantitative analysis of fluorescence in the worm nervous system. The authors demonstrate the performance of the software on two test cases: an over-expressed transgene and an endogenously-tagged synaptic protein. They also quantify the ability of their approach to detect differences between wild-type and mutant fluorescence phenotypes as a demonstration of its robustness. Appealingly, the imaging approach is fundamentally 2D rather than the common 1D approach used by multiple groups when analyzing nerve cord synaptic signals. And the software includes multiple approaches to quantification of colocalization of two fluorescence signals from two-color imaging. Overall, this is an impressive effort that could improve the quality of quantitative nervous system imaging in C. elegans if widely adopted. I have a few questions and clarifications for the authors to address but no major issues with this manuscript or with the details of the methodology.

1. It is helpful and informative that the authors demonstrated the analysis routines applied to both high signal-to-noise (SNR) data and low SNR data. I was left wondering how well this analysis works as a function of the number of independent worms included in the data set. For instance, do you need significantly more animals in the low SNR case to get the same statistical power as in the high SNR case? I’m not sure how many biological replicates went into the data in Figure 1, but Figure 2 includes what I am guessing are the number of independent images (worms) that went into the ROI count analysis in the various genotypes. Is the 30-40 number fairly typical for the types of analysis performed by this software? Side note - ideally the Fig 2 legend should comment on what the numbers within the bars are referring to. Some commentary on the number of animals needed for reliable estimates of puncta counts, sizes, intensities, etc… could be valuable for future users of this approach.

2. For the colocalization analysis, it was great to see the authors compare different types of colocalization metrics. Figure 4 was a bit confusing and the legends could be improved by including some more information. For instance, at the bottom of Fig 4A, what is the significance of showing the ROI perimeters for the >40% threshold case but not showing them for the <40% threshold case where colocalization is scored as true? And this figure would have been more informative if the reader could see how the overall analysis of colocalization turned out using different colocalization definitions and thresholds rather than just showing the behavior at one or two representative puncta. Its hard to get a feel for what the final colocalization conclusions are or how the aggregate measures of colocalization would change when switching between methods.

3. One minor point on the efforts to look at the impact of noise (and the plots in the Figure 4 supplement). I do not have a feel for how to compare the relative noise levels used in the four noise classes. This would have been more informative if the authors plotted the final output (the normalized correlation metric for example) as a function of noise intensity for each of the four classes of noise rather than at a particular (and perhaps arbitrary or undefined) level of applied background noise. The noise could be increased from 0 to 100% where 100% means the same total intensity is contributed to the cropped image from the noise as from the signal. In this context, the relative robustness of the different metrics to noise class may be more obvious.

4. The results section opens with a somewhat abstract discussion of the SNR for the over-expressed and endogenous reporters, and I would have followed better if there was an associated figure with a zoom-in of some representative ROIs (more like what is shown in Figure 4 but emphasizing SNR, Bradley’s method, etc…).

Minor:

What is the gray dashed line in Figure 1D (local background maybe)? This should be mentioned in the legend or annotated in the figure. Also, the scale bar is more useful if positioned in the image rather than off to the side where the figure annotations/keys are sitting.

Reviewer #2: The manuscript introduces a no-code, user friendly GUI-based MATLAB application tailored for the detection and analysis of fluorescent puncta in C. elegans nerve cord images. The software employs adaptive local means thresholding and watershed segmentation, offering an alternative to traditional global or 1D projection-based methods. Through comprehensive validation—including comparisons with ground truth annotations, metrics, and robustness checks—WormSNAP is shown to outperform existing approaches. It balances accuracy, scalability, and usability, and is designed to be accessible to non-programmers.

Strengths of the work include its methodological rigor, careful benchmarking against other tools, and strong emphasis on user-friendliness, reproducibility, and open access. The GUI design, scalability testing, and inclusion of detailed outputs make WormSNAP well-suited for broad adoption in the C. elegans research community. Limitations include: the tool is constrained to 2D two-channel analysis and relies on a manual cropping pre-processing step, which precludes fully automated pipelines. Additionally, while the validation datasets are well chosen, broader testing across different imaging platforms and organisms would help generalize its utility. Nonetheless, this manuscript presents a meaningful contribution to computational neurobiology and image analysis in model organisms.

This paper appears to be well suited for publication in PLOS Computational Biology, and I support its publication pending revision based on the following comments.

Here are suggested major changes:

1. The local mean thresholding is generally a great segmentation method, but it often fails when there is another signal in the same FOV after cropping, and many times, and in many cases, cropping alone cannot eliminate other nearby signals. Also, as presented in Figure 2G, you may find it hard to find a very good sensitivity before testing different sensitivity, which is not ideal for larger dataset analysis (and it is generally faster to manually count puncta if the dataset is small). I suggest the author consider trying an image filter (perhaps the 2D LoG filter) to highlight those puncta-like structures, and then perform the local mean thresholding. You may find that a precise crop is not necessary after the LoG (or other filter) image enhancement. If this combined method proves to be more effective than local mean thresholding only, please add it to your paper and software. If not, please simply reply with your result to this review. I am also wondering if adding a LoG filter may make the puncta segmentation on typical epifluorescence images possible.

Here are suggested minor changes:

1. In introduction, please add a few paragraphs about why detection and characterization of puncta in C. elegans is an important topic in biology.

2. In Fig. 1Di, I personally cannot tell whether the groundtruth annotation is correct, especially the fourth puncta. Can the author either change the round green circle around each puncta to arrows pointing to them, or manually adjust the gamma of the image to make the figure itself look higher contrast for the audience (but explain this in figure caption), or combine these two methods? The author can also try other methods for readability of the original puncta.

3. In section of thresholding and segmentation algorithm, please add more explanation to the last formula.

4. Local thresholding often has different performance on areas that are much denser or much sparser than the average signal density. Can author provide some thoughts on this?

5. The first paragraph in Results: WormSNAP has been developed to process cropped and straightened images of up to two fluorescent reporters, hereafter simply termed crops. The term of straightened images is not very commonly used in C. elegans community. Please add a quick explanation to it, if the author thinks it is better to add.

6. Section GUI Pipeline, second paragraph, typo Figure 4A.

7. Figure 3d caption typo, Figure 4 Supp C. Please check the rest of your paper for those kinds of typos, as I see many of these kinds of problems.

8. Figure 4: Adding numerical values of Manders’ coefficients and overlap ratios near each example would clarify metric sensitivity. The term “threshold” might confuse readers (intensity threshold vs. overlap percentage). Adding clarification in the caption would help.

9. Please provide some test images of puncta in your GitHub page so people can quickly replicate and verify the segmentation process with WormSnap.

Reviewer #3: In this manuscript, the authors provide and describe a MatLab-based software, named WormSNAP, to detect and count fluorescent puncta. The software sets regions of interest (ROI) using adjustable local means thresholding, which the authors show is more robust than previously described methods including global thresholding to define synaptic puncta. The authors showed that software was able to detect the known defects of puncta distributions with severe or mild phenotypes.

The authors also demonstrate the ability to directly visualize the ROIs from two different channels and overlay them. These ROIs can then be used to quantify colocalization of puncta from the separate channels and calculate the Pearson’s Correlation Coefficient, Mander’s Coefficients, and a newly described ROI Overlap Ratio, which determines the fraction of ROIs that are overlapping.

Overall, the software and GIU appear to be user friendly and will aid in determining ROIs that indicate fluorescent puncta seen in confocal images, as well as quantifying these ROIs. I have several comments/suggestions which I hope authors could address before it is published.

Major Comments:

1. The major limitation of WormSNAP is that, due to how the software exclude noises and background signals such as intestinal autofluorescence, the neuron types that are compatible with the software depend largely on their anatomy; they must be relatively simple, and the puncta must be arranged in a straight line. This may greatly reduce the number of researchers who can benefit from WormSNAP. Can the authors show if WormSNAP can quantify synapses of the HSN neuron in supplemental figure 1? If the use of WormSNAP is limited to the neurons in the nerve cord, the authors should describe it as a major limitation in the discussion section.

2. Throughout the manuscript, the authors appear to carefully avoid using C. elegans gene and protein names, especially in the Figures. While the authors implied that this software has the potential to be useful for researchers using other models, this is very much optimized for examining synapses of C. elegans as the name of the software suggests. Not describing the details of the neurons used, the proteins visualized, and the mutant backgrounds used will cause confusion to the main target readers of this manuscript.

3. Related to the above comment, the authors used different genetic backgrounds to represent ‘severe’ and ‘mild’ phenotypes in Figure 2 without defining them in the main text or in the figure, which is very confusing. Also, the phenotypes of nrx-1 and syd-1 mutants that the authors used in Figure 2A-D are qualitatively different and hence not appropriate to be described as mild and severe phenotypes, respectively. Furthermore, the authors compared these mutants only with wildtype. Does the WormSNAP detect the difference between severe and mild mutants?

4. The authors demonstrate, in a step-by-step manner, the robustness of the WormSNAP, in identifying fluorescent synaptic puncta, however, some quantifications lack representative images for viewers to determine the accuracy of the ROI.

a. Figure 1E-G: Only images of overexpressed CLA-1::GFP is shown but the representative endogenous NRX-1:Skylan-S with ground truth annotation, peak finder, global threshold, and local means threshold are not shown.

b. Figure 3D-G: Raw image of C2 and with C2 overlays would be helpful.

c. Figure Supplemental 4A, B: visualization of both fluorescent channels as a visual representation of the Pearson’s correlation,

5. Showing the WormSNAP’s ability to detect and segment larger ROIs from a fluorophore with lower signal-to-noise ratio (Ex. Red fluorescent protein tagged with a synaptic vesicle protein) would provide users a greater understanding of the usability of the GUI. This is shown indirectly in Figure 3F and 3G but is not directly addressed.

Minor Comments:

1. Please follow C. elegans genetic nomenclature. For example, the gene names should be italicized, and genes on the same chromones should not have semi-colon in between them.

2. Figure 1D: Please include the image with no ROI overlay. Some ‘groundtruth’ puncta look almost like ‘noise’ by the authors’ definition.

3. Regarding the GUI validation, can the authors suggest a recommended PC specification? Alternatively, the authors could indicate the PC specs they used in this work.

4. Figure 1E-F: Graph titles are confusing, consider using Sensitivity (TPR) and Specificity (TNR) instead of using “/”

5. Supplemental Figure 4C: The names of noises do not match between the Figure and the main text.

6. There are lots of incorrect figure citations including:

a. GUI Pipeline – “After ROIs are detected in all available channels, users proceed to the Results tab (Supp Figure 3C) to visualize the ROIs and configure output settings. Using the ROI Display panel, users can easily view ROIs. When viewing 2-channel images, users can view ROIs of each channel (Figure 3C)…” Figure 3C does not show ROIs of each channel.

b. Colocalization metrics – “Noise is a major consideration when selecting a correlation metric22. We generated synthetic two-channel datasets based on Channel 1(CLA-1::GFP) of the Overexpressed Synaptic Fluorophore CTRL crops with four different noise types designed to mimic typical confocal microscopy issues: Thermal noise, Shot noise, Salt and Pepper noise, and Speckle noise (Supp Figure 3C).” Should cite Supp Figure 4C.

7. Figure panels should be cited in order as first cited. For example, Supp Figure 3F appears before Supp Figure 3D and E.

8. Comparisons of PCC, MC, and ROI Overlap Ratios using actual images of synaptic vesicles and active zone proteins would provide further insight.

9. It was unclear what the preprocessing step of “drawing crops” as stated in the discussion entailed, whether it only included straightening of images to include synaptic/axonal regions, or additional work was required.

10. Comparisons to other simple software or macros (in addition to WormPsyQi) would be helpful in the discussion. Other open-source software that uses free software, such as ImageJ/FIJI has been developed and published to detect and determine synaptic puncta size, fluorescence intensity, and co-localizations.

https://www.sciencedirect.com/science/article/pii/S266723752400239X

https://www.micropublication.org/journals/biology/micropub-biology-001003

**Have the authors made all data and (if applicable) computational code underlying the findings in their manuscript fully available?**

Reviewer #1: Yes

Reviewer #2: Yes

Reviewer #3: Yes

PLOS authors have the option to publish the peer review history of their article (what does this mean? ). If published, this will include your full peer review and any attached files.

**Do you want your identity to be public for this peer review?** For information about this choice, including consent withdrawal, please see our Privacy Policy .

Reviewer #1: No

Reviewer #2: No

Reviewer #3: No

**Figure resubmission:**
---

## [Decision Letter · Decision Letter 1]

20 Oct 2025

Dear Dr. Kurshan,

We are pleased to inform you that your manuscript 'WormSNAP: A software for fast, accurate, and unbiased detection of fluorescent puncta in C. elegans' has been provisionally accepted for publication in PLOS Computational Biology.

Best regards,

Adriana San Miguel

Academic Editor

PLOS Computational Biology

Feilim Mac Gabhann

Editor-in-Chief

PLOS Computational Biology

Dear Dr. Kurshan,

I am happy to share that your revised article is now accepted for publication in PLOS Computational Biology. As you can see from the reviewer's comments, most concerns were addressed appropriately. I ask you to please address the minor comments by Reviewer 2 during the final manuscript check, which are mainly concerned with formatting or typos.

Best regards,

Adriana San Miguel

Reviewer's Responses to Questions

**Comments to the Authors:**

Reviewer #1: The authors have done an excellent job in responding to my questions and comments. I think the manuscript is in good shape and have no further concerns.

Reviewer #2: In this revision, all my previous concerns are resolved by the author. But one problem of this edition is the low resolution of all figures, which might be caused by their editing method or the submission system. Before I see a good quality of figures, I cannot make further decisions.

Except the figure resolution problem, below are minor changes:

1. Line 39-41, please add a few references to support your statement.

2. Figure 3A caption, please confirm you are referring to Local Area Thresholding or your most mentioned Local Means Thresholding.

3. Line 66, reference 8 and 12 format.

4. Line 550, typo

5. In your reply to my previous question 9, you said that A pre-Analyzed Dataset has now been added to

Github. However, I cannot find it on github.

Reviewer #3: The authors thoroughly addressed the reviewers' comments.

**Have the authors made all data and (if applicable) computational code underlying the findings in their manuscript fully available?**

Reviewer #1: Yes

Reviewer #2: Yes

Reviewer #3: Yes

PLOS authors have the option to publish the peer review history of their article (what does this mean? ). If published, this will include your full peer review and any attached files.

**Do you want your identity to be public for this peer review?** For information about this choice, including consent withdrawal, please see our Privacy Policy .

Reviewer #1: No

Reviewer #2: No

Reviewer #3: No

---

## [Editor Report · Acceptance letter]

PCOMPBIOL-D-25-00888R1

WormSNAP: A software for fast, accurate, and unbiased detection of fluorescent puncta in C. elegans

Dear Dr Kurshan,

I am pleased to inform you that your manuscript has been formally accepted for publication in PLOS Computational Biology. Your manuscript is now with our production department and you will be notified of the publication date in due course.

With kind regards,

Zsofia Freund
